# Playing Nondeterministic Games through Planning with a Learned Model

## Abstract

The MuZero algorithm is known for achieving high-level performance on traditional zero-sum two-player games of perfect information such as chess, Go, and shogi, as well as visual, non-zero sum, single-player environments such as the Atari suite. Despite lacking a perfect simulator and employing a learned model of environmental dynamics, MuZero produces game-playing agents comparable to its predecessor, AlphaZero. However, the current implementation of MuZero is restricted only to deterministic environments. This paper presents Nondeterministic MuZero (NDMZ), an extension of MuZero for nondeterministic, two-player, zero-sum games of perfect information. Borrowing from Nondeterministic Monte Carlo Tree Search and the theory of extensive-form games, NDMZ formalizes chance as a player in the game and incorporates the chance player into the MuZero network architecture and tree search. Experiments show that NDMZ is capable of learning effective strategies and an accurate model of the game.

## 1 Introduction

While the AlphaZero algorithm achieved superhuman performance in a variety of challenging domains, it relies upon a perfect simulation of the environment dynamics to perform precision planning. MuZero, the newest member of the AlphaZero family, combines the advantages of planning with a learned model of its environment, allowing it to tackle problems such as the Atari suite without the advantage of a simulator. This paper presents Nondeterministic MuZero (NDMZ), an extension of MuZero to stochastic, two-player, zero-sum games of perfect information. In it, we formalize the element of chance as a player in the game, determine a policy for the chance player via interaction with the environment, and augment the tree search to allow for chance actions.

As with MuZero, NDMZ is trained end-to-end in terms of policy and value. However, NDMZ aims to learn two additional quantities: the player identity policy and the chance player policy. With the assumption of perfect information, we change the MuZero architecture to allow agents to determine whose turn it is to move and when a chance action should occur. The agent learns the fixed distribution of chance actions for any given state, allowing it to model the effects of chance in the larger context of environmental dynamics. Finally, we introduce new node classes to the Monte Carlo Tree Search, allowing the search to accommodate chance in a flexible manner. Our experiments using Nannon, a simplified variant of backgammon, show that NDMZ approaches AlphaZero in terms of performance and attains a high degree of dynamics accuracy.

## 2 Prior Work

### 2.1 Early Tree Search

The classic search algorithm for perfect information games with chance events is expectiminimax, which augments the well-known minimax algorithm with the addition of of chance nodes (Mitchie, 1966; Maschler et al., 2013). Chance nodes assume the expected value of a random event taking place, taking a weighted average of each of its children, based on the probability of reaching the child, rather than the max or min. Chance nodes, min nodes, and max nodes are interleaved depending on the game being played. The *-minimax algorithm extends the alpha-beta tree pruning strategy to games with chance nodes (Ballard, 1983; Knuth & Moore, 1975).

## 2.2 MODEL-FREE REINFORCEMENT LEARNING

A notable success of model-free reinforcement learning for nondeterministic games is TD-Gammon, an adaptation of Sutton & Barto (2018)'s TD-Lambda algorithm to the domain of backgammon (Tesauro, 1995). Using temporal difference learning, a network is trained to produce the estimated value of a state. When it is time to make a move, each legal move is used to produce a successor state, and the move with the maximum expected outcome is chosen. Agents produced by TD-Gammon were able to compete with world-champion backgammon players, although it was argued that the stochastic nature of backgammon assisted the learning process (Pollack & Blair, 1996).

## 2.3 MONTE CARLO TREE SEARCH

The Monte Carlo Tree Search (MCTS) algorithm attracted interest after its success with challenging deterministic problems such as the game Go, and it has been adapted for stochastic games in an number of different ways (Coulom, 2006; Browne et al., 2012). A version of MCTS by Van Lishout et al. (2007) called called McGammon (Monte Carlo Backgammon) was capable of finding expert moves in a simplified backgammon game. At a chance node, the next move is always chosen randomly; at choice nodes, the Upper Confidence bound applied to Trees (UCT) algorithm is used in the selection phase to choose player actions (Kocsis & Szepesvári, 2006).

The McGammon approach was adapted by Yen et al. (2014) to the game of Chinese Dark Chess and formalized as Nondeterministic Monte Carlo Tree Search (NMCTS). There are two node types in NMCTS: deterministic state nodes and nondeterministic state nodes. Nondeterministic state nodes contain at least one inner node; each inner node has its own visit count, win count, and subtree. Upon reaching a nondeterministic state node, an inner node is chosen using "roulette wheel selection" based on the probability distribution of possible resulting states. NMCTS was designed to suit the nature of Chinese Dark Chess, in which both types of state nodes may exist on the same level of the search tree; this is in contrast to backgammon, in which homogeneous layers of chance nodes and choice nodes are predictably interleaved.

Alternative MCTS approaches have been adapted for densely stochastic games, wherein the the branching factor at chance nodes is so great that successor states at chance nodes are unlikely to ever be resampled. These include Double Progressive Widening and Monte Carlo *-Minimax Search (Chaslot et al.; Lanctot et al., 2013). In this paper, however, we are concerned with games that are not densely stochastic.

## 2.4 ALPHAZERO AND MUZERO

MCTS served as the foundation of the AlphaGo and AlphaZero algorithms, the latter of which attained superhuman performance in the deterministic domains of chess, Go, and shogi (Silver et al., 2016; b;a). AlphaZero substitutes neural network activations for random rollouts to produce policy and value evaluations, employs the Polynomial Upper Confidence Trees (PUCT) algorithm during search, and trains solely via self-play reinforcement learning (Rosin, 2011). Hsueh et al. (2018) adapted AlphaZero for a simplified, solved version of the nondeterministic game of Chinese Dark Chess, finding that AlphaZero is capable of attaining optimal play in a stochastic setting.

A notable limitation of AlphaZero is that the agent relies upon a perfect simulator in order to perform its lookahead search. The MuZero algorithm frees AlphaZero of this restriction, creating a model of environmental dynamics as well as game-playing strategies through interactions with the environment and self-play (Schrittwieser et al., 2019). Until recently, the state of the art in complex single-player domains such as Atari games involved model-free reinforcement learning approaches such as Q-learning and the family of algorithms that followed (Mnih et al., 2015; Çalışır & Pehlivanoğlu, 2019). Despite the burden of learning enviornmental dynamics, MuZero leverages its precision-planning capabilities to achieve an extremely strong degree of performance on these problems.

Given parameters $\theta$ and a state representation at timestep $k$, AlphaZero employs a single neural network $f_\theta$ with a policy head and a value head: $\boldsymbol{p}^k, v^k = f_\theta(s^k)$. In contrast, the MuZero model employs three networks for representation, prediction, and dynamics. A set of observations is passed into the representation function $h_\theta$ to produce the initial hidden state: $s^0 = h_\theta((o_1, \ldots, o_t))$. The dynamics function recurrently takes a previous hidden state and an action image to produce an

immediate reward and the next hidden state: $r^k, s^k = g_\theta(s^{k-1}, a^k)$. Any hidden state produced in such fashion may be passed into the prediction function $f_\theta$ to produce a policy vector and scalar value: $\boldsymbol{p}^k, v^k = f_\theta(s^k)$.

With this model, the agent may search over future trajectories $a^1, ..., a^k$ using past observations $o_1, ..., o_t$; in principle any MDP planning algorithm may be used given the rewards and states produced by the dynamics function. MuZero uses a MCTS algorithm similar to AlphaZero's search, wherein the representation function is used to generate the hidden state for the root node. Each child node then has its own hidden state, produced by the dynamics function, and prior probability and value, produced by the prediction function. These values are used in the tree search to select an action $a_{t+1} \sim \pi_t$ via PUCT.

# 3 NONDETERMINISTIC MUZERO

## 3.1 DEFINITIONS

We consider here an extension of the MuZero algorithm for two-player, zero-sum games of perfect information with chance events. Following Lanctot et al. (2019), we will characterize such games as extensive-form games with the tuple $\langle \mathbb{N}, \mathbb{A}, \mathbb{H}, \mathbb{Z}, u, \iota, \mathbb{S} \rangle$, wherein

- $\mathbb{N} = \{1, 2, c\}$ includes the two rival players as well as the special **chance player** $c$;
- $\mathbb{A}$ is the finite set of actions that players may take;
- $\mathbb{H}$ is the set of histories, or sequence of actions taken from the start of the game;
- $\mathbb{Z} \subseteq \mathbb{H}$ is the subset of terminal histories;
- $u : \mathbb{Z} \to \Delta_u^n \subseteq \mathbb{R}^n$, where $\Delta_u = [u_{min}, u_{max}]$, is the utility function assigning each player a utility at terminal states;
- $\iota : \mathbb{H} \to \mathbb{N}$ is the player identity function, such that $\iota(h)$ identifies the player to act at history $h$;
- $\mathbb{S}$ is the set of states, in which $\mathbb{S}$ is a partition of $\mathbb{H}$ such that each state $s \in \mathbb{S}$ contains histories $h \in s$ that cannot be distinguished by $\iota(s) = \iota(h)$ where $h \in s$.
- The legal actions available at state $s$ are denoted as $\mathbb{A}(s) \subseteq \mathbb{A}$;
- Zero-sum games are characterized such that $\forall z \in \mathbb{Z}, \sum_{i \in \mathbb{N}} u_i(z) = 0$; and
- A game of perfect information is one with only one history per state: $\forall s \in \mathbb{S}, |s| = 1$.

A **chance node** (or chance event) is a history $h$ such that $\iota(h) = c$. A **policy** $\pi : \mathbb{S} \to \Delta(\mathbb{A}(s))$, where $\Delta(\mathbb{X})$ represents the set of probability distributions over $\mathbb{X}$, describes agent behavior. An agent acts by sampling an action from its policy: $a \sim \pi$. A **deterministic** policy is one where the distribution over actions has probability 1 on one action and zero on others. A policy that is not necessarily deterministic is called **stochastic**. The chance player always plays with a fixed, stochastic policy $\pi_{\text{chance}}$.

A **transition function** $\mathbb{T} : \mathbb{S} \times \mathbb{A} \to \Delta(\mathbb{S})$ defines a probability distribution over successor states $s'$ when choosing action $a$ from state $s$. Because states are simply sequences of previous actions, a transition function can equivalently be represented using intermediate chance nodes between the histories of the predecessor and successor states $h \in s$ and $h' \in s'$. The transition function is then determined by $\pi_{\text{chance}}$.

As we are concerned with perfect information games, we shall use $s$ interchangeably with $h$ throughout this text, such that $s$ may refer to the single history $h$ contained within it. For ease of notation we refer to the chance policy $\pi_{\text{chance}}$ as $\psi$.

## 3.2 MODIFICATIONS

First, we shall extend $\mathbb{N}$ such that $\mathbb{N} = \{1, 2, c, d\}$ includes the new **terminal player** $d$. The terminal player is to act when a terminal state is reached, such that $\forall z \in \mathbb{Z}, \iota(z) = d$. We shall refer to nonchance, nonterminal players as **choice players**. Let us characterize the full action set $\mathbb{A}$ as the

union of the choice player action set $\mathbb{A}_\pi$, the chance player action set $\mathbb{A}_\psi$, and the terminal player action set $\mathbb{A}_d$ as follows: $\mathbb{A} = \mathbb{A}_\pi \cup \mathbb{A}_\psi \cup \mathbb{A}_d$. Let us introduce the special **no-op action**, which is produced when it is not a player's turn to move, such that $\forall n \in \mathbb{N}$ and $\forall s \in \mathbb{S}, \mathbb{A}_n(s) = \{\text{no-op}\}$ when $\iota(s) \neq n$. The action set $\mathbb{A}_d$ of the terminal player contains the no-op action as well as the **end action**, which represents ending the game: $\mathbb{A}_d = \{\text{no-op, end}\}$. We can see that for any given state, the set of legal chance actions and the set of legal choice actions are both subsets of the set of all actions: $\forall s \in \mathbb{S}, \mathbb{A}_\psi(s) \subseteq \mathbb{A}$, and $\forall s \in \mathbb{S}, \mathbb{A}_\pi(s) \subseteq \mathbb{A}$.

To provide input $s^{k-1}, a^k$ to the dynamics function $g_\theta$, the original MuZero encodes the action performed as an image and stacks it with the current hidden state. As with any choice player action, we may then encode any chance action $a \in \mathbb{A}_\psi$ taken from $\psi(s)$ as an image and stack it with a previous hidden state to produce input for $g_\theta$.

For an agent to properly model a stochastic environment, it must determine when chance events occur as well as the distributions of their effects. The problem then is to learn an approximation of the player identity function $\iota(s)$ as well as the chance player policy $\psi(s)$. We then add two quantities that our model, conditioned by parameters $\theta$, must predict: the **chance policy** $c^k \approx \psi_{t+k}$ and the **player identity policy** $i^k \approx \iota_{t+k}$ in addition to the quantities already present in MuZero: the policy $p^k \approx \pi_{t+k}$ (which we now term the **choice policy**), value $v^k \approx z_{t+k}$, and reward $r^k \approx u_{t+k}$.

Two modifications are made to the MuZero network architecture. The prediction function now produces the choice policy, chance policy, and value: $p^k, c^k, v^k = f_\theta(s^k)$, while the dynamics function now produces the reward, next hidden state, and player identity policy: $r^k, s^k, i^k = g_\theta(s^{k-1}, a)$, where the length of $i^k = |\mathbb{N}|$. We use the no-op action as a training target for the choice policy and chance policy when it is not that player's turn to move.

Adding the L2 regularization term, we define our loss function as follows:

$$l_t(\theta) = \sum_{k=0}^{K} l^r(u_{t+k}, r_t^k) + l^v(z_{t+k}, v_t^k) + l^p(\pi_{t+k}, p_t^k) + l^c(\psi_{t+k}, c_t^k) + l^i(\iota_{t+k}, i_t^k) + c||\theta||^2 \quad (1)$$

### 3.3 Tree Search

Equipped with these new quantities, we wish to modify our MCTS to accommodate chance nodes. To flexibly capture the dynamics of stochastic games, the model must determine who is to play before either making a player action, making a chance action, or terminating the search. We introduce the **identity layer**, a layer of nodes that we interleave with **action layers**. The identity layer employs our $\iota$ approximation to determine the direction of the search.

Let us declare the root node to be a **choice node**. As in MuZero, the root hidden state $s^0$ is generated by passing observations into the representation function such that $s^0 = h_\theta((o_1, \ldots, o_t))$. We then derive the priors of the root node's children by feeding the root hidden state into the prediction function: $p^0, _-, _- = f_\theta(s^0)$; here, the chance policy and value may be ignored. Following MuZero, the children of the root node are taken from the set of legal actions derived from the true environment state, $\mathbb{A}_\pi(s^t)$, and assigned edges corresponding to those actions; likewise, we select among the children of a choice node with PUCT during tree search.

The children of choice nodes are **identity nodes**. Upon expansion, the reward $r^k$, next hidden state $s^k$, and identity policy $i^k$ of an identity node is generated by applying the dynamics function to the hidden state of its parent $s^{k-1}$ along with the image of its action: $r^k, s^k, i^k = g_\theta(s^{k-1}, a)$. The prediction function is then applied to the next hidden state $s^k$, yielding the choice policy, chance policy, and value: $p^k, c^k, v^k = f_\theta(s^k)$. An identity node has a number of children equal to $|\mathbb{N}|$; thus, in our example each identity node has four children corresponding to $\{1, 2, c, d\}$. A child is selected from an identity node using roulette wheel selection proportional to $i^k$. The children from the choice player edges $\{1, 2\}$ become choice nodes, the child from the chance edge $c$ becomes a **chance node**, and the child from the terminal edge $d$ becomes a **terminal node**.

Each **chance node** and nonroot choice node inherits the hidden state $s^k$ from its parent identity node and has a number of children equal to $|\mathbb{A}|$. (We do not assume prior knowledge of $|\mathbb{A}_\pi|$ or $|\mathbb{A}_\psi|$; so long as we begin with $|\mathbb{A}|$, these are learned and approximated through training.) A choice node also inherits from its parent the value $v^k$, and its children take their priors from the choice policy $p^k$. A chance node inherits the chance policy $c^k$ from its parent and uses this to generate the priors of its

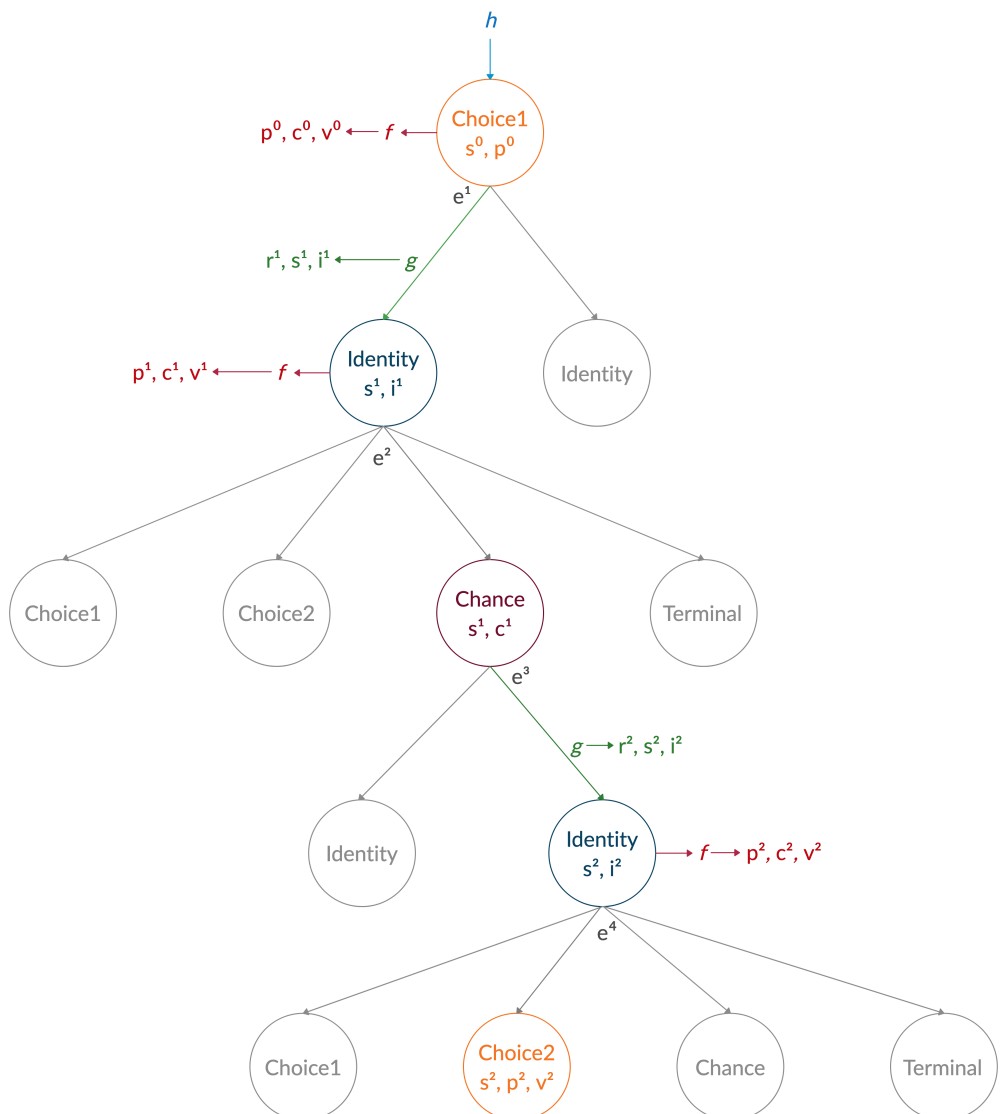

Figure 1: Illustration of NDMZ tree search. $e^k$ represents the edge between nodes.

children; during search, the child of a chance node is also chosen using roulette wheel selection. All children of chance nodes and nonroot choice nodes are identity nodes.

In contrast with MuZero, backup is only performed when the leaf node is either a choice node or a terminal node. First, let us consider the case when the leaf node is a choice node. We will refer to identity nodes with a choice node parent as **result nodes**. We only employ PUCT during search when selecting among result nodes and use roulette wheel selection otherwise. Because of this, we only need add or subtract the value obtained from the leaf node to the result nodes found in the visit path. If the edge leading to the leaf node is the same as the edge leading to the parent of the result node, we add the value of the leaf node to the total value of the result node; otherwise, we subtract. If the leaf node is a terminal node, we start with the parent of the leaf node and search up the tree until we find the most recent choice node. We note the value and the edge leading to this most recent choice node, return to the leaf node, and proceed with backup as described above.

Pseudocode for a version of the algorithm applicable to board games lacking immediate rewards is given in full in Appendix A.

## 4 EXPERIMENTS

### 4.1 NANNON

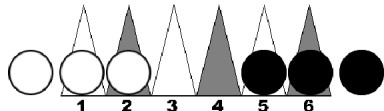

For testing, benchmarking, and evaluation, we used Nannon, a simplified variant of backgammon that is solvable through value iteration (Pollack, 2005). Nannon is properly considered as a parameterized set of games, such that the number of board points, checkers per player, and sides of dice may be changed, influencing the complexity of the game. It is played by rolling a single die. There is no stacking, and only one checker per point is allowed; however, two adjacent pieces of the same color form a block. The number of possible board states may be found with the following equation, where $n$ is the number of points on the board and $k$ is the number of checkers per player:

$$\sum_{i=0}^{k}\sum_{j=0}^{k}\binom{n}{i}\binom{n-i}{j}(k+1-i)(k+1-j) \tag{2}$$

We chose two game configurations, both with a six-sided die: six points on the board with three checkers per player (6-3-6) and twelve points on the board with five checkers per player (12-5-6). 6-3-6 Nannon has 2,530 states, and 12-5-6 Nannon has 1,780,776 states. The first game configuration has some surprising subtleties despite the small number of total states; the second, while much more complex, is still far more tractable than backgammon. Each configuration was solved using value iteration, yielding a lookup table with which the optimal policy could be derived (Bellman, 1957). This policy is used to guide the actions of the **optimal player**; by pitting agents produced by AlphaZero and NDMZ against the optimal player, we obtain a more objective estimate of their performance.

### 4.2 EXECUTION AND HYPERPARAMETERS

The NDMZ algorithm was executed as follows. The Nannon game engine code was written in C++ using the OpenSpiel framework, while the algorithm was written in Python using Tensorflow 2.4. Dense multi-layer perceptrons were used instead of convolutional resnets. Models for $f_\theta$, $g_\theta$, and $h_\theta$ were initialized with two hidden layers each of 256 weights, using SELU activation and LeCun Normal initialization (Klambauer et al., 2017). Experiments were carried out on a Kubernetes cluster with Ray using 500 CPU workers for evaluation and 300 CPU workers for self-play; a single T4 GPU was used for training.

One hundred rounds of self-play, training, and evaluation were performed. For both AlphaZero and NDMZ, the models run for three hundred games of self-play per round, while the tree search performs one hundred simulations per move. The 6-3-6 run used a replay buffer limit of 4,000 games, while the 12-5-6 run used a limit of 1,000, as games on the larger board are roughly four times longer; for both configurations, the replay buffer carries about 100,000 examples total. For MuZero, samples are pulled at random from randomly chosen games, and a $K$ value of six is used, where K is the number of hypothetical steps in which the network is unrolled and trained via backpropagation through time. Around 600,000 examples are then seen per epoch for NDMZ and 100,000 for AlphaZero.

Training for both algorithms employs an Adam optimizer with a learning rate of 0.001 and a batch size of 512. We did not scale the hidden states to the same range as the action input, nor did we scale the loss by $\frac{1}{K}$ or $\frac{1}{2}$. For the first twenty rounds of the trial, five epochs of training are performed; afterwards, only a single epoch is done. Evaluation of the AlphaZero and NDMZ agents is performed once before training begins and after every training iteration, consisting of 1,000 rounds of play versus the optimal player and 1,000 rounds of play versus an agent that makes moves at random (500 games of each color). For comparison with model-free algorithms, we ran the TD-Lambda algorithm on the same Nannon configurations. Three hundred games of self-play are performed per round, while training occurs with a learning rate of 0.001, $\lambda = 0.7$, and $\gamma = 1.0$.

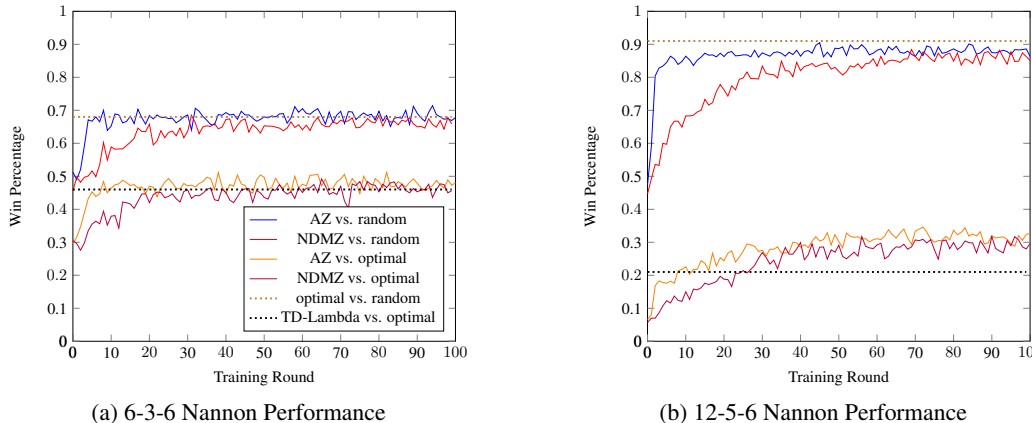

(a) 6-3-6 Nannon Performance

(b) 12-5-6 Nannon Performance

Figure 2: Performance evaluation results of agents produced by the NDMZ and AlphaZero (AZ) algorithms in the game of Nannon throughout the duration of a training run, compared with the average results of a trained TD-Lambda agent.

### 4.3 DYNAMICS EVALUATION

Tests were designed to provide more practical insight into the ability of the networks to capture the dynamics of the game. After each training iteration, five hundred games are played with random moves and two tests are performed.

For the first test, the game trajectories are analyzed in the following manner: For every position at timestep $t$ reached in every game, the choice policy output $\boldsymbol{p}^t$ of $f_\theta$ is compared with the actual legal moves for that position: $\mathbb{A}_\pi(s_{\text{env}}^t)$. If the choice policy ranks any illegal move with greater probability than a legal move, the position is marked a failure; otherwise, it is a success. This proceeds recurrently for $K$ times, using the dynamics network $g_\theta$ along with the move actually played to produce a new hidden state that is fed into $f_\theta$, obtaining $\boldsymbol{p}^{t+k}$ for each $k$ in $1, \ldots, K$. The actual legal moves for these future states $s^{t+1}, \ldots, s^{t+K}$ are compared with the recurrently produced policies, and the results for all positions are averaged. We call this the **top move dynamics test**.

The second test proceeds along similar lines as the first; however, the position is marked a failure if any illegal move is given greater than uniform probability. For example, if $\boldsymbol{p}^t$ has length 4 and an illegal move has probability $> 0.25$, the test for that position is a failure; otherwise it is a success. This we call the **uniform dynamics test**. Each test evaluates around 60,000 positions for 6-3-6 Nannon and 240,000 positions for 12-5-6 Nannon.

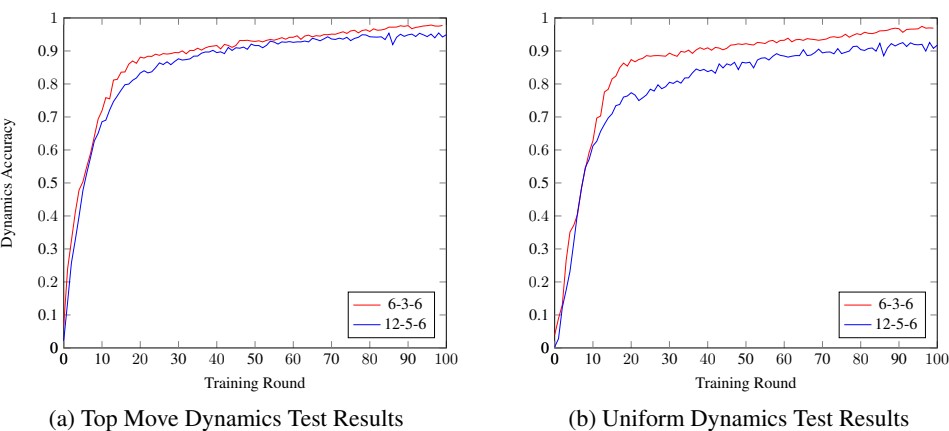

(a) Top Move Dynamics Test Results

(b) Uniform Dynamics Test Results

Figure 3: Results of dynamics tests as described in Section 4.3.

## 4.4    RESULTS AND CONCLUSION

We observe in Figure 2 that, for both configurations, AlphaZero experiences a very sharp and immediate increase in performance against the random player; the performance of NDMZ, on the other hand, exhibits a gradual curve before reaching a plateau of near-convergence with the AlphaZero agent. This is similar to some of the results shown in Figure 2 of Schrittwieser et al. (2019), where we see that the Elo performance of the MuZero agent eventually converges with that of the AlphaZero agent in the domains of chess and shogi. Moreover, we see that both AlphaZero and NDMZ exceed the performance of the TD-Lambda agent by a significant margin in the 12-5-6 configuration. The dynamics experiments in Figure 3 show that NDMZ agents can predict valid moves with a high degree of accuracy even several moves deep in the search. By comparing the two graphs, we also observe that the performance of the agent improves as it better learns to model its environment.

From these results, we may conclude that NDMZ is capable of learning effective strategies and a useful model of the game. Future work could entail testing NDMZ on games with more unorthodox chance node interleavings, such as Chinese Dark Chess, and extending the algorithm to accommodate more complex single-player environments such as the Atari suite.

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

# A APPENDIX

---

**Algorithm** Nondeterministic MuZero Tree Search

---

1: **function** NDMZSEARCH($s^{\text{env}}$)                         ▷ $s^{\text{env}}$ is the raw environment state
2:     create root node $w_0$
3:     EXPANDROOT($w_0, s^{\text{env}}$)                    ▷ adds children and hidden state to root
4:     **while** within computational budget **do**
5:         $w_L \leftarrow$ TREESELECT($w_0$)                              ▷ returns a leaf node
6:         **if** $K(w_L) =$ choice or $K(w_L) =$ terminal **then**          ▷ $K$ is the kind of node
7:             BACKUP($w_L, v$)
8:     $\mathbb{W} \leftarrow$ children of $w_0$                       ▷ $N$ is the number of visits
9:     $w_1 \leftarrow$ sample a child $w'$ from $\mathbb{W}$ proportional to $\frac{N(w')^{1/\tau}}{\sum_{b \in \mathbb{W}} N(b)^{1/\tau}}$          ▷ $\tau$: temperature param
10:     **return** $A(w_1)$

11:
12: **function** EXPANDROOT($w_0, s^{\text{env}}$)
13:     $S(w_0) \leftarrow h_\theta(s^{\text{env}})$                ▷ representation network assigns hidden state $S$ to root.
14:     $K(w_0) \leftarrow$ choice                              ▷ root is choice node
15:     $\boldsymbol{p}, \_, \_ \leftarrow f_\theta(S(w_0))$               ▷ prediction network: choice policy logits
16:     $\boldsymbol{p} \leftarrow$ masked softmax of $(\boldsymbol{p}, \mathbb{A}_\pi(s^{\text{env}}))$             ▷ scale for legal actions only
17:     **for** $e \in \mathbb{A}_\pi(s^{\text{env}})$ **do**          ▷ for each edge $e$ taken from the set of legal actions
18:         add child $w'$ with                      ▷ $N$: visit count; $W$: total value;
19:         $N(w') \leftarrow 0; W(w') \leftarrow 0; Q(w') \leftarrow 0;$      ▷ $Q$: mean value; $P$: prior probability;
20:         and $P(w') \leftarrow \boldsymbol{p}_a; E(w') \leftarrow e$              ▷ $E$: edge leading to node

21:
22: **function** BACKUP($w$)
23:     **if** $K(w) =$ terminal **then**                              ▷ if $w$ is a terminal node
24:         $n \leftarrow w$
25:         **while** $K(n) \neq$ choice **do**
26:             $n \leftarrow$ parent of $n$          ▷ search up the tree until we find the closest choice node
27:         $e \leftarrow E(n)$                              ▷ player identity of the choice node
28:         $v \leftarrow V(n)$                              ▷ value from the choice node
29:     **else**
30:         $e \leftarrow E(w)$                              ▷ player identity of the leaf node
31:         $v \leftarrow V(w)$                              ▷ value from leaf node
32:     **while** $w$ is not none **do**
33:         $j \leftarrow$ parent of $w$
34:         **if** $K(w) =$ identity and $K(j) =$ choice **then**              ▷ if $w$ is a result node
35:             **if** $E(j) = e$ **then**                    ▷ if the edge leading to $j$ is the same
36:                 $W(w) \leftarrow W(w) + v$                              ▷ add to total value
37:             **else**
38:                 $W(w) \leftarrow W(w) - v$                    ▷ subtract from total value
39:             $N(w) \leftarrow N(w) + 1$                              ▷ increment visit count
40:             $Q(w) \leftarrow \frac{W(w)}{N(w)}$                              ▷ adjust mean value
41:         $w \leftarrow$ parent of $w$

---

42: **function** TREESELECT($w$)
43:  **while** $K(w)$ is not terminal **do**
44:   **if** $w$ not expanded **then**                $\triangleright$ $w$ is a leaf node
45:    **return** EXPAND($w$)         $\triangleright$ adds child nodes, hidden state, and value
46:   $\mathbb{W} \leftarrow$ children of $w$
47:   **if** $K(w) =$ chance or $K(w) =$ identity **then**      $\triangleright$ chance or identity node
48:    $w \leftarrow$ sample $w'$ from $\mathbb{W}$ weighted by $P(w')$     $\triangleright$ roulette wheel selection
49:   **else if** $K(w) =$ choice **then**            $\triangleright$ choice node
50:    $w \leftarrow \underset{w' \in \mathcal{W}}{\arg\max} \left( Q(w') + c_{\text{puct}} P(w') \frac{\sqrt{\sum_{b \in \mathcal{W}} N(b)}}{1 + N(w')} \right)$     $\triangleright$ PUCT
51:  **return** $w$
52:
53: **function** EXPAND($w$)
54:  $j \leftarrow$ parent of $w$
55:  **if** $K(j) =$ choice or $K(j) =$ chance **then**   $\triangleright$ if $w$ is the child of a choice or chance node
56:   $K(w) \leftarrow$ identity             $\triangleright$ $w$ is an identity node
57:   $\_, S(w), \boldsymbol{p} \leftarrow g_\theta(S(j), A(w))$      $\triangleright$ for board games, we ignore the reward
58:   $O(w), C(w), V(w) \leftarrow f_\theta(S(w))$     $\triangleright$ choice policy, chance policy, and value
59:   $\mathbb{C} \leftarrow \mathbb{N}$               $\triangleright$ $\mathbb{C}$ is the set of child edges
60:  **else if** $K(j) =$ identity **then**        $\triangleright$ if $w$ is the child of an identity node
61:   $S(w) \leftarrow S(j)$        $\triangleright$ $w$ always inherits the hidden state of its parent
62:   **if** $A(w) = c$ **then**        $\triangleright$ if the edge leading to $w$ is the chance edge $c$
63:    $K(w) \leftarrow$ chance           $\triangleright$ $w$ is a chance node
64:    $\boldsymbol{p} \leftarrow C(j)$           $\triangleright$ $\boldsymbol{p}$ is the chance policy vector
65:    $\mathbb{C} \leftarrow \mathbb{A}$            $\triangleright$ $\mathbb{A}$ is the full action set
66:   **else if** $A(w) = d$ **then**       $\triangleright$ if the edge leading to $w$ is the terminal edge $d$
67:    $K(w) \leftarrow$ terminal          $\triangleright$ $w$ is a terminal node
68:    $\mathbb{C} \leftarrow \{\}$          $\triangleright$ terminal nodes have no children
69:   **else**
70:    $K(w) \leftarrow$ choice           $\triangleright$ $w$ is a choice node
71:    $\boldsymbol{p} \leftarrow O(j)$           $\triangleright$ $\boldsymbol{p}$ is the choice policy vector
72:    $V(w) \leftarrow V(j)$          $\triangleright$ $w$ inherits value from parent
73:    $\mathbb{C} \leftarrow \mathbb{A}$
74:  **for** $e \in \mathbb{C}$ **do**          $\triangleright$ for each edge in the set of child edges
75:   add child $w'$ with
76:   $N(w') \leftarrow 0; W(w') \leftarrow 0; Q(w') \leftarrow 0;$
77:   and $P(w') \leftarrow \boldsymbol{p}_e; E(w') \leftarrow e$   $\triangleright$ assign prior to child from $\boldsymbol{p}$ along with edge $e$
78:  **return** $w$

