# OpenReview forum: "Playing Nondeterministic Games through Planning with a Learned Model"
_ICLR.cc/2021/Conference — Reject_

### Official Review · AnonReviewer3 · 2020-10-23
**ICLR 2021 Conference Paper2090 AnonReviewer3**

**Rating:** 7
**Confidence:** 1

**Review:**

This paper presents an algorithm NDMZ that extends MuZero to non-deterministic, two-player, zero-sum games of perfect information. The new algorithm borrows the idea from non-deterministic MCTS and the theory of extensive-form games. The empirical studies show a competitive performance of MuZero agains AlphaGoZero, despite MuZero lacks a perfect simulator the game.

Comments:

This paper is generally well-written and clear. The empirical results demonstrate that NDMZ can achieve a similar performance as AlphaGoZero, which is very interesting provided that MuZero does not get access to a perfect game simulator.

1. In Fig 2. with 12-5-6 Nannon, it shows that it takes AlphaGoZero a few rounds to learn and after 100 rounds of training, it does not perform well against the optimal policy (with a winning rate about 30%), which implies that AlphaGoZero does not converge to an optimal policy. The reviewer is not familiar with AlphaGoZero's performance in these games and is wondering whether this is expected.

---

> ### Author Response · Authors · 2020-11-24
> **Performance question**
>
> We thank you for your review. Both algorithms indeed did not reach the same performance as the optimal policy. Different durations and parameter configurations were tried, but these represent some of the most successful results. This is not yet fully understood but could be an interesting area of further work.

---

### Official Review · AnonReviewer4 · 2020-10-26
**Empirical evaluation can be improved**

**Rating:** 5
**Confidence:** 4

**Review:**

Summary
The paper extends MuZero for nondeterministic domains (NDMZ). Compared to MuZero NDMZ also learns a function that determines who is to act (player 1,2 or chance) and a distribution of chance outcomes. This makes it possible to employ MCTS search adjusted to handle nondeterministic nodes on top of a tree constructed by NDMZ's neural nets.


Strong points
* NDMZ is a straightforward extension of MuZero, therefore it might be interesting to a broad audience.
* The model is clearly described.

Weak points
* Empirical evaluation could provide more insights about the usefulness of NDMZ.


Recommendation

I recommend rejecting the paper on the basis of insufficient empirical evaluation.


Questions

* Could you provide broader context for the presented results? How do other learning approaches perform on the selected games?

* While Nannon 12-5-6 is still a relatively small game there is a significant gap in win rate of NDMZ against optimal player. Is this a good result?


Possible improvements

* Discussion of results is very brief.

* The most important question not answered by current evaluation is: "Does proposed search help over model free methods with similar number of trainable parameters?" e.g. how far from optimum would DQN (or something similar) be in these games?

* Second important missing experiment is: "Does more search help? How does performance scale with more MCTS iterations?" One of promises of search based systems is that they can make use of more "thinking" time during play. It should be easy to test whether NDMZ has this property too.

* Final (obvious) improvement would be to test NDMZ on an environment that is large enough that it can't be strongly solved (e.g. Backgammon) and show the importance of search compared to model free approaches there. While scaling up might be nontrivial it would significantly improve the message of the paper.


Minor

* For MuZero, a K value of six is used, and samples are pulled at random from randomly chosen games before being unrolled K times and trained via backpropagation through time. ... It is strange to refer to K before explaining what it is.

* I assume that in Section 4 AZ refers to AZ run with the exact search tree from Sec 3.3 however I believe that it isn't explicitly stated.

* called called -> called



===== Post rebuttal update =====

I like the addition of TD Gammon like baseline and as a result of that I slightly increased my score. However please add more details about that baseline (e.g. did it have a comparable number of parameters to NDMZ?). You can also use performance of just the policy head (as a model free agent) to show what is the contribution of search.
Also showing how performance scales with search would still be a valuable experiment.

---

> ### Author Response · Authors · 2020-11-24
> **Empirical evaluation**
>
> Thank you for your review. We had earlier performed an additional set of experiments using the TD-Lambda algorithm, which is the classic model-free algorithm used by the TD-Gammon backgammon engine. We found that TD-Lambda performed about equally well as AlphaZero against the random player, but lagged behind both AlphaZero and NDMZ when pitted against the optimal player. We include this result in the updated paper.
>
> There is indeed a gap in the win rate of NDMZ against the optimal player, but this is true of AlphaZero as well. The primary point of interest is that while AlphaZero quickly reaches a plateau of performance, NDMZ gradually converges to a similar plateau. This shows that NDMZ succeeds at modeling dynamics in a useful way to attain comparable performance with AlphaZero. However, it remains an interesting question why the gap exists, which could be a topic of future work.
>
> To your second question, we feel that a deeper exploration of search in these algorithms is an issue worthy of its own paper and outside of the scope of this work. Your final concerns echo those of Reviewer 1 and Reviewer 5, both of whom felt that more experiments on more complex domains were required.
>
> Minor issues:
>
> The K value has been clarified: “a K value of six is used, where K is the number of hypothetical steps in which the network is unrolled and trained via backpropagation through time.”
>
> In Figure 2, NDMZ refers to the exact search tree from Sec 3.3. AZ refers to running AlphaZero with chance nodes.

---

### Official Review · AnonReviewer2 · 2020-10-29
**Promising initial work, that needs some writing revision.**

**Rating:** 6
**Confidence:** 4

**Review:**

Summary:

This paper presents NDMZ, an extension of the MuZero algorithm to handle games with chance moves. In order to do this, chance nodes are added to MCTS, and add a few additional prediction targets for the network, such as predicting when a chance event will occur, so that it can be simulated in MCTS.

Reasons for score:

I am split on this paper, on the one hand I think this is an interesting direction, and the results, even in a toy domain like Annon, show promising results. Specially Figure 3, where we can see that the learned forward model gets more and more accurate over time, which is encouraging. On the other hand, I felt that some of the new formalization is a bit clumsy. The paper could also use some work on making the descriptions more formal, clear and concise (specially Section 3.3, which includes many vague and lengthy descriptions that can be made shorter and more concise with a few definitions).



Additional feedback:

- page 3: on the definition of state, you mention that histories belonging to a state cannot be distinguished by which player is next. But probably another constraint is that they cannot be distinguished either by the set of available actions, right?
- pages 3-4: I find the addition of player "d" and the no-op action unnecessary, and just making the formalization clumsy. This is not needed any most other formalization of games with chance events. I do not see they extending the expressiveness of the formalization in any way, and just add unnecessary complexity. Why do you even need to train the policy of an agent when it's not its turn to move? Can't you simply skip its update, rather than making it learn to predict an arbitrary "no-op" value that will never be used, but will affect the network weights?

---

> ### Author Response · Authors · 2020-11-24
> **Writing revisions**
>
> We are thankful for your review. We agree that simplifications could be made; some of the lengthier paragraphs came as a response to feedback from other readers who wanted more explanation. The purpose of the “d” player is primarily practical as a means of terminating the tree search. We decided to include it in the formalism as it is in the set of outcomes for the identity policy. We agree that the no-op value could be omitted, but since it was used in experiments, the value should be retained in this version of the work.

---

### Official Review · AnonReviewer1 · 2020-10-30
**Insufficient contribution and experiments.**

**Rating:** 4
**Confidence:** 4

**Review:**

This paper proposes NDMZ, which extends the previous MuZero algorithm to stochastic two-layer zero-sum games of perfect information. NDMZ formalize chance as a player (chance player) and introduces two additional quantities: the player identity policy and the chance player policy. NDMZ also introduce new node classes to MCTS, which allows it to accommodate chance.

One major weakness of the paper is there is its lack of novelty/contribution. The core idea of interleaving chance nodes with choice nodes to model stochastic environment in tree search is not brand new. The main contribution of this paper is the integration of such idea into the specific MuZero tree search framework. This might be OK if the authors could present strong enough experimental results.

However, this brings up a second main weakness of the paper, which is lack of sufficient experimental justification. The proposed NDMZ is only evaluated on Nannon, a simplified version of backgammon, which is not sufficient. I would suggest the authors to evaluate the algorithm on at least 5~10 Atari games. Although the Atari environment is known to be a deterministic, it becomes stochastic when there is only a limited horizon of past observed frames. In addition, just like MuZero, the proposed NDMZ should also be adapted to single-player tasks like Atari games. Therefore, under this setting, Atari could be used to simulate the stochastic environment for the current purpose. In addition, it would also be helpful to further evaluate it on the more complicated tasks such as Chinese Dark Chess, as suggested by the authors.

Finally, the dynamic evaluation (including both top-move dynamic test and uniform dynamics test) only evaluate the accuracy of the learned dynamics by only examining the chance of selecting illegal move. This seems to be just one very restricted perspective of dynamics evaluation. I’m wondering whether there should be other more comprehensive ways to do this?

---

> ### Author Response · Authors · 2020-11-24
> **More experiments needed**
>
> We appreciate your review and agree that an expanded scope and fuller suite of experimental results would improve this work. The contribution here is primarily technical and is restricted to two-player games of perfect information, so we understand your concerns, which are similar to those of Reviewer 5. Future work would include evaluations on complex single-player domains such as the Atari suite. Regarding your final point, the original MuZero paper primarily focuses on performance measures, but we wanted to evaluate the ability of such algorithms to model dynamics as well. The tests here were designed to be more intuitive than other measures such as training error: we would suspect that a person has not learned the dynamics of chess if they reliably make illegal moves. However, alternative measures of evaluation could be interesting future work.

---

### Official Review · AnonReviewer5 · 2020-11-05
**Clear and enjoyable paper, but difficult to motivate**

**Rating:** 3
**Confidence:** 4

**Review:**

This paper introduces NDMZ, short for nondeterministic MuZero, a deep reinforcement learning algorithm for model-based RL that doesn't use the rules of the game to perform search. The paper's contribution is mostly focused on describing how to construct the algorithm, and experimental results are provided at the end. A good analogy is that of a player that must play a (physical) board game by not only making decisions, but also acting out the game: producing random events, such as die rolls, and moving pieces on the board.

Overall, I enjoyed this paper and thought it was quite clear, but it does not feel substantial enough for a conference publication. The main contribution is the detail of how to implement stochasticity in a MuZero architecture. While interesting, there's relatively little discussion of why these are the right choices, why this is particularly challenging, or in fact what value it adds compared to existing algorithms.

My main concern with the paper can be summarized as: What is this work's impact? What is the key premise that makes it a reasonable line of work? It seems that AlphaZero is just as well equipped to deal with games (where a simulator *is* available; the restriction imposed here is artificial). Demonstrating that a deep learning system can model stochastic events isn't too surprising either. I would have liked to see more discussion and empirical support arguing why this particular work brings new insights to deep RL. The easiest way to do so is to demonstrate a problem where NDMZ or MZ outperforms AlphaZero, or where a model isn't available, or is too cumbersome to be used.

A minor issue also concerns the presentation. The algorithm is described in terms of 'chance policy' and 'identity policy', but I would call these just a transition model. To present them as additional players is a little surprising.

Minor points:
- PUCT: spell out the name
- Why did you omit the L2 term in Eqn 1?
- Why is the root node necessarily a choice node?

---

> ### Author Response · Authors · 2020-11-24
> **More Impactful Experiments**
>
> We thank you for your review. This work presents an extension of the MuZero algorithm to an admittedly restricted domain: two-player zero-sum games of perfect information. In the original paper, the authors show that MuZero converges with or slightly exceeds AlphaZero in terms of performance in the domains of chess, Go, and shogi. Our paper shows analogous results for NDMZ in a simplified backgammon game. However, one striking accomplishment of the MuZero paper was its performance on the Atari suite, in which it exceeded the R2D2 algorithm. As Reviewer 1 suggests, it may be that an adaptation of this work to Atari domains or others of similar complexity would make this work more salient. Given the evident power of MuZero in deterministic environments, a more broad extension to stochastic environments could prove useful for solving difficult problems other than games.
>
> -The ‘chance policy’ and ‘identity policy’ terms were used to follow the formalisms of extensive-form games; they are equivalent to a transition model.
> -Fixed the PUCT
> -There was a typesetting issue, fixed.
> -At the root node, the identity is known and the agent must make a choice.

---

### Decision · Program_Chairs · 2021-01-07
**Final Decision**

**Decision:**

Reject

**Comment:**

There is a pretty good consensus that this paper should not be accepted at ICLR. The reviewers do not seem think that extending MuZero to non-deterministic MuZero constitutes a significant advance.  Three reviewers give clear rejects with scores (3, 4, 5) all with good confidence (4).  A fourth reviewer gave a score of 6, i.e., borderline accept.  While the fifth reviewer recommends, he does not seem to be very confident and did not step in to champion the paper.  The program committee decided that the paper in its current form does not meet the acceptance bar.